# Hypersolvers: Toward Fast Continuous-Depth Models

**Michael Poli**[*]
KAIST, DiffEqML
poli_m@kaist.ac.kr

**Stefano Massaroli**[*]
The University of Tokyo, DiffEqML
massaroli@robot.t-u-tokyo.ac.jp

**Atsushi Yamashita**
The University of Tokyo
yamashita@robot.t.u-tokyo.ac.jp

**Hajime Asama**
The University of Tokyo
asama@robot.t.u-tokyo.ac.jp

**Jinkyoo Park**
KAIST
jinkyoo.park@kaist.ac.kr

## Abstract

The infinite–depth paradigm pioneered by Neural ODEs has launched a renaissance in the search for novel dynamical system–inspired deep learning primitives; however, their utilization in problems of non–trivial size has often proved impossible due to poor computational scalability. This work paves the way for scalable Neural ODEs with *time–to–prediction* comparable to traditional discrete networks. We introduce `hypersolvers`, neural networks designed to solve ODEs with low overhead and theoretical guarantees on accuracy. The synergistic combination of `hypersolvers` and Neural ODEs allows for cheap inference and unlocks a new frontier for practical application of continuous–depth models. Experimental evaluations on standard benchmarks, such as sampling for *continuous normalizing flows*, reveal consistent pareto efficiency over classical numerical methods.

## 1 Introduction

The framework of *neural ordinary differential equations* (Neural ODEs) (Chen et al., 2018) has reinvigorated research in continuous deep learning (Zhang et al., 2014), offering new system–theoretic perspectives on neural network architecture design (Greydanus et al., 2019; Bai et al., 2019; Poli et al., 2019; Cranmer et al., 2020) and generative modeling (Grathwohl et al., 2018; Yang et al., 2019). Despite the successes, Neural ODEs have been met with skepticism, as these models are often slow in both training and inference due to heavy numerical solver overheads. These issues are further exacerbated by applications which require extremely accurate numerical solutions to the differential equations, such as physics–inspired neural networks (Raissi et al., 2019) and continuous normalizing flows (CNFs) (Chen et al., 2018).

Common knowledge within the field is that these models appear too slow in their current form for meaningful large-scale or embedded applications. Several attempts have been made to either directly or indirectly address some of these limitations, such as redefining the forward pass as a root finding problem (Bai et al., 2019), introducing *ad hoc* regularization terms (Finlay et al., 2020; Massaroli et al., 2020a) and augmenting the state to reduce stiffness of the solutions (Dupont et al., 2019; Massaroli et al., 2020b).

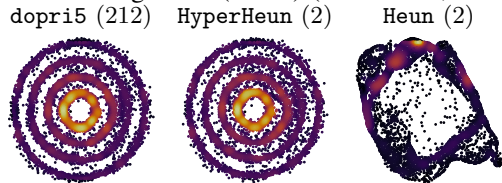

Figure 1: Hypersolvers for density estimation via *continuous normalizing flows*: `dopri5` inference accuracy is achieved with 100x speedup.

Unfortunately, these approaches either give up on the Neural ODE formulation altogether, do not reduce computation overhead sufficiently or introduce additional memory requirements. Although there is no shortage of works utilizing Neural ODEs in forecasting or classification tasks (Yıldız et al., 2019;

---

[*]Equal contribution. Author order was decided by flipping a coin.

Jia & Benson, 2019; Kidger et al., 2020), current state–of–the–art is limited to offline applications with no constraints on inference time. In particular, high–potential application domains for Neural ODEs such as control and prediction often deal with tight requirements on inference speed and computation e.g robotics (Hester, 2013) that are not currently within reach. For example, a generic state–of–the–art convolutional Neural ODE takes at least an order of magnitude[2] longer to infer the label of a *single* MNIST image. This inefficiency results in inference passes far too slow for real–time applications.

| Method | NFEs | Local Error |
|---|---|---|
| $p$-th order solver | $\mathcal{O}(pK)$ | $\mathcal{O}(\epsilon^{p+1})$ |
| adaptive–step solver | – | $\mathcal{O}(\tilde{\epsilon}^{p+1})$ |
| Euler hypersolver | $\mathcal{O}(K)$ | $\mathcal{O}(\delta\epsilon^2)$ |
| $p$-th order hypersolver | $\mathcal{O}(pK)$ | $\mathcal{O}(\delta\epsilon^{p+1})$ |

Figure 2: Asymptotic complexity comparison. Number of function evaluations (NFEs) needed to compute $K$ solver's steps. $\epsilon$ is the step size, $\tilde{\epsilon}$ is the max step size of adaptive solvers, $\delta \ll 1$ is correlated to the `hypersolver` training results.

● **Model–solver synergy**   The interplay between Neural ODEs and numerical solvers has largely been overlooked as research on model variants has been predominant, often treating solver choice as a simple hyper–parameter to be tuned based on empirical observations. Here, we argue for the importance of computational scalability outside of specific Neural ODE architectural modifications, and highlight the synergistic combination of model–solver to be a likely candidate for unlocking the full potential of continuous–depth models. Namely, this work attempts to alleviate computational overheads by introducing the paradigm of Neural ODE hypersolvers; these auxiliary neural networks are trained to solve the initial value problem (IVP) emerging from the forward pass of continuous–depth models. `Hypersolvers` improve on the computation–correctness trade–off provided by traditional numerical solvers, enabling fast and arbitrarily accurate solutions during inference.

● **Pareto efficiency**   The trade–off between solution accuracy and computation is one of the oldest and best–studied topics in the numerics literature (Butcher, 2016) and was mentioned in the seminal work (Chen et al., 2018) as a feature of continuous models. Traditional approaches shift additional compute resources into improved accuracy via higher–order adaptive–step methods (Prince & Dormand, 1981). For the most part, the computation–accuracy pareto front determined by traditional methods has been treated as optimal, allowing practitioners its traversal with different solver choices. We provide theoretical and practical results in support of the pareto efficiency of `hypersolvers`, measured with respect to both *number of function evaluations* (NFEs) as well as standard indicators of algorithmic complexity. Fig. 2 provides a comparison of `hypersolvers` and traditional methods.

● **Inference speed**   By leveraging Hypersolved Neural ODEs, we obtain significant speedups on common benchmarks for continuous–depth models. In image classification tasks, inference is sped up by *at least* one order of magnitude. Additionally, the proposed approach is capable of solving *continuous normalizing flow* (CNF) (Chen et al., 2018; Grathwohl et al., 2018) sampling in few steps with little–to–no degradation of the sample quality as shown in Fig. 4.1. Moving beyond computational advantages at inference time, the proposed framework is compatible with *continual learning* (Parisi et al., 2019) or *adversarial learning* (Ganin et al., 2016) techniques where model and `hypersolver` are co–designed and jointly optimized. Sec. 6 provides an overview of this peculiar interplay.

## 2   Background: Continuous-Depth Models

We start by introducing necessary background on Neural ODE and numerical integration methods.

**Neural ODEs**   We consider the following general Neural ODE formulation (Massaroli et al., 2020b)

$$\begin{cases} \dot{\mathbf{z}} = f_{\theta(s)}(s, \mathbf{x}, \mathbf{z}(s)) \\ \mathbf{z}(0) = h_x(\mathbf{x}) \qquad\qquad s \in \mathcal{S} \\ \hat{\mathbf{y}}(s) = h_y(\mathbf{z}(s)) \end{cases} \tag{1}$$

with input $\mathbf{x} \in \mathbb{R}^{n_x}$, output $\hat{\mathbf{y}} \in \mathbb{R}^{n_y}$, *hidden* state $\mathbf{z} \in \mathbb{R}^{n_z}$ and $\mathcal{S}$ is a compact subset of $\mathbb{R}$. Here $f_{\theta(s)}$ is a neural network, parametrized by $\theta(s)$ in some functional space. We equip the Neural ODE

with input and output mappings $h_x : \mathbb{R}^{n_x} \to \mathbb{R}^{n_z}, h_y : \mathbb{R}^{n_z} \to \mathbb{R}^{n_y}$ which are kept linear as to avoid a collapse of the dynamics into a non-necessary map as discussed in (Massaroli et al., 2020b).

**Solving the ODE**   Without any loss of generality, let $\mathcal{S} := [0, S]$ ($S \in \mathbb{R}^+$). The inference of Neural ODEs is carried out by solving the *initial value problem* (IVP) (1), i.e.

$$\hat{\mathbf{y}}(S) = h_y \left( h_x(\mathbf{x}) + \int_{\mathcal{S}} f_{\theta(\tau)}(\tau, \mathbf{x}, \mathbf{z}(\tau)) \mathrm{d}\tau \right)$$

Due to the nonlinearities of $f_{\theta(s)}$, this solution cannot be defined in closed–form and, thus, a numerical solution should be obtained by iterating some predetermined ODE solver. Let us divide $\mathcal{S}$ in $K$ equally–spaced intervals $[s_k, s_{k+1}]$ such that for all $k \in \mathbb{N}_{<K}$  $s_{k+1} - s_k = S/K := \epsilon \in \mathbb{R}^+$. The numerical approximation of the IVP solution in $\mathcal{S}$ can be computed by iterating

$$\begin{cases} \mathbf{z}_{k+1} = \mathbf{z}_k + \epsilon \psi(s_k, \mathbf{x}, \mathbf{z}_k) \\ \quad \mathbf{z}_0 = h_x(\mathbf{x}) \qquad\qquad\quad k = 0, 1, \ldots, K-1 \\ \quad \hat{\mathbf{y}}_k = h_y(\mathbf{z}_k) \end{cases} \qquad (2)$$

where $\psi$ is a function performing the state update.

**Numerical methods**   ODE solvers differ in how this map $\psi$ is constructed[3]. In example, the Euler method is realized by setting $\psi(\mathbf{x}, s_k, \mathbf{z}_k) := f_{\theta(s_k)}(\mathbf{x}, s_k, \mathbf{z}_k)$. Note that, *higher–order* solvers compute $\psi(\mathbf{x}, s_k, \mathbf{z}_k)$ iteratively in $p$ steps where $p$ denotes the order of the solver. For example, in a $p$-th order Runge-Kutta (RK) (Runge, 1895) method $\psi$ is computed as

$$\begin{aligned} \mathbf{r}_i &= f_{\theta(s_k)}(s_k + \mathbf{c}_i\epsilon, \mathbf{x}, \mathbf{z}_k + \tilde{\mathbf{z}}_k^i) \quad i = 1, \ldots, p \\ \tilde{\mathbf{z}}_k^i &= \epsilon \sum\nolimits_{j=1}^{p} \mathbf{a}_{ij}\mathbf{r}_j \qquad\qquad i = 1, \ldots, p \\ \psi &= \sum\nolimits_{j=1}^{p} \mathbf{b}_j\mathbf{r}_j \end{aligned} \qquad (3)$$

where $\mathbf{a} \in \mathbb{R}^{p \times p}$, $\mathbf{b} \in \mathbb{R}^p$, $\mathbf{c} \in \mathbb{R}^p$ fully characterize the method. Hence, the integration of a neural ODE in $\mathcal{S}$ with a RK solver is $\mathcal{O}(pK)$ in memory efficiency and time complexity. On the other hand, *adaptive–step* solvers, *e.g.* the popular Dormand–Prince 5(4) (`dopri5`) have no explicit upper bounds in memory and time efficiency. This is especially critical as in many practical applications, a requirement for maximum memory consumption and/or inference time must be satisfied.

**Common metrics**   In classic numerical analysis, two type of metrics are often defined, i.e. the *local truncation error* $e_k$

$$e_k := \|\mathbf{z}(s_{k+1}) - \mathbf{z}(s_k) - \epsilon\psi(s_k, \mathbf{x}, \mathbf{z}(s_k))\|_2,$$

representing the error accumulated in a single step, and the *global truncation error* $\mathcal{E}_k$ is

$$\mathcal{E}_k = \|\mathbf{z}(s_k) - \mathbf{z}_k\|_2,$$

i.e. the error accumulated in the first $k$ steps. Note that for a $p$-th order solver $e_k = \mathcal{O}(\epsilon^{p+1})$ and $\mathcal{E}_k = \mathcal{O}(\epsilon^p)$ (Butcher, 2016).

# 3 ● `Hypersolvers` **for Neural ODEs**

`Hypersolvers` offers a computational framework for the interplay between Neural ODEs and their numerical solver. The core idea behind `hypersolvers` is to introduce an additional neural network $g_\omega$ to approximate the higher–order terms of a given solver, greatly increasing its accuracy while preserving the computational and memory efficiency. The simplest instance of `Hypersolved` Neural ODEs is based on Euler scheme:

$$\begin{cases} \mathbf{z}_{k+1} = \mathbf{z}_k + \epsilon f_{\theta(s_k)}(s_k, \mathbf{x}, \mathbf{z}_k) + \epsilon^2 g_\omega(\epsilon, s_k, \mathbf{x}, \mathbf{z}_k) \\ \quad \mathbf{z}_0 = h_x(\mathbf{x}) \qquad\qquad\qquad\qquad\qquad\quad k = 0, 1, \ldots, K-1 \\ \quad \hat{\mathbf{y}}_k = h_y(\mathbf{z}_k) \end{cases} \qquad (4)$$

where $g_\omega$ is a neural network approximating the second–order term of the Euler method. The derivation of the *Euler hypersolver* comes naturally from the following. Let $\mathbf{z}(s_k)$ be the true solution of (1) at $s_k \in \mathcal{S}$ and let $\epsilon > 0$ such that $s_k + \epsilon \in \mathcal{S}$. From the Taylor expansion of the solution around $s_k$, i.e.

$$\mathbf{z}(s_k + \epsilon) = \mathbf{z}(s_k) + \epsilon\dot{\mathbf{z}}(s_k) + \frac{1}{2}\epsilon^2\ddot{\mathbf{z}}(s_k) + \mathcal{O}(\epsilon^3)$$

$$\approx \mathbf{z}(s_k) + \epsilon f_\theta(s_k, \mathbf{x}, \mathbf{z}(s_k))$$

we deduce that the classic Euler scheme corresponds to the first–order truncation of the above. The Euler `hypersolver`, instead, aims at approximating the second–order term, reducing the local truncation error of the overall scheme, while avoiding to compute and store further evaluations of $f_{\theta(s)}$, as required by higher order schemes, e.g. RK methods.

## 3.1 General formulation

A general formulation of `Hypersolved` Neural ODEs can be obtained extending (2). If we assume $\psi$ to be the update step of a $p$-th order solver, then the general $p$-th order `Hypersolved` Neural ODE is defined as

$$
\begin{aligned}
\mathbf{z}_{k+1} &= \mathbf{z}_k + \epsilon\overbrace{\psi(s_k, \mathbf{x}, \mathbf{z}_k)}^{\text{solver step}} + \epsilon^{p+1}\overbrace{g_\omega(\epsilon, s_k, \mathbf{x}, \mathbf{z}_k)}^{\text{hypersolver net}} \quad & k = 0, 1, \dots, K-1 \\
\mathbf{z}_0 &= h_x(\mathbf{x}) \\
\hat{\mathbf{y}}_k &= h_y(\mathbf{z}_k)
\end{aligned}
\tag{5}
$$

**Software implementation**   We implemented `hypersolver` variants of common low–order explicit ODE solvers, designed for compatibility with the `TorchDyn` (Poli et al., 2020) library[4]. The Appendix further includes a PyTorch (Paszke et al., 2017) module implementation.

## 3.2 Training `hypersolvers`

Assume to have available the *exact* solution of the Neural ODE evaluated at the mesh points $s_k$, practically obtained through an adaptive–step solver set up with low tolerances. With these solution checkpoints we construct the training set for the DE solver with tuples:

$$\{(s_k, \mathbf{z}(s_k))\}_{k\in\mathbb{N}_{\leq K}}$$

According to the introduced metrics $e_k$ and $\mathcal{E}_k$, we introduce two types of loss functions aimed at improving each of the metrics.

**Residual fitting**   We first start by defining the *residual* of the solver (2)

$$\mathcal{R}(s_k, \mathbf{z}(s_k), \mathbf{z}(s_{k+1})) = \frac{1}{\epsilon^{p+1}}\left[\mathbf{z}(s_{k+1}) - \mathbf{z}(s_k) - \epsilon\psi(s_k, \mathbf{x}, \mathbf{z}(s_k))\right] \tag{6}$$

which correspond to a scaled local truncation error without the neural correction term $g_\omega$. Then, we can consider a loss measuring the discrepancy between the residual terms and the output of $g_\omega$:

$$\ell = \frac{1}{K}\sum_{k=0}^{K-1}\|\mathcal{R}(s_k, \mathbf{z}(s_k), \mathbf{z}(s_{k+1})) - g_\theta(\epsilon, s_k, \mathbf{x}, \mathbf{z}(s_k))\|_2$$

If the `hypersolver` is trained to minimize $\ell_{\texttt{local}}$, the following holds:

**Theorem 1** (Hypersolver Local Truncation Error). *If $g_\omega$ is a $\mathcal{O}(\delta)$ approximator of $\mathcal{R}$, i.e.*

$$\forall k \in \mathbb{N}_{\leq K} \quad \|\mathcal{R}(s_k, \mathbf{z}(s_k), \mathbf{z}(s_{k+1}) - g_\theta(\epsilon, s_k, \mathbf{x}, \mathbf{z}(s_k))\|_2 \leq \mathcal{O}(\delta),$$

*then, the local truncation error $e_k$ of the* `hypersolver` *is $\mathcal{O}(\delta\epsilon^{p+1})$.*

The proof and further theoretical insights are reported in the Appendix.

`https://github.com/DiffEqML/diffeqml-research/tree/master/hypersolver`

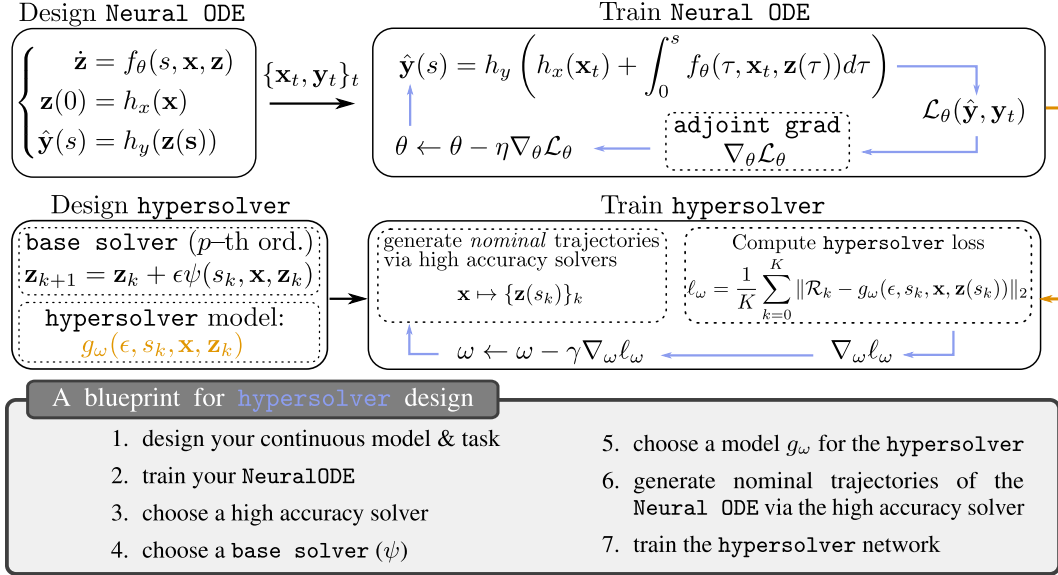

**Trajectory fitting** The second type of `hypersolvers` training aims at containing the global truncation error by minimizing the difference between the exact and approximated solutions in the whole depth domain $\mathcal{S}$, i.e.

$$L = \sum_{k=1}^{K} \|\mathbf{z}(s_k) - \mathbf{z}_k\|_2$$

It should be noted that *trajectory* and *residual fitting* can be combined into a single loss term, depending on the application.

## 4 Experimental Evaluation

The evaluation protocol is designed to measure `hypersolver` pareto efficiency, inference time speedups and generalizability across base solvers. We consider the following general benchmarks for Neural ODEs: standard image classification (Dupont et al., 2019; Massaroli et al., 2020b) and density estimation with continuous normalizing flows (CNFs) (Chen et al., 2018; Grathwohl et al., 2018).

### 4.1 Image Classification

We train standard convolutional Neural ODEs with input–layer augmentation (Massaroli et al., 2020b) on MNIST and CIFAR10 datasets. Following this initial optimization step, 2–layer convolutional *Euler hypersolvers*, `HyperEuler`, (4) are trained by residual fitting (6) on 10 epochs of the *training* dataset with solution mesh length set to $K = 10$. As ground–truth labels, we utilize the solutions obtained via `dopri5` with absolute and relative tolerances set to $10^{-4}$ on the same data. The objective of this first task is to show that `hypersolvers` retain their pareto efficiency when applied in high–dimensional data regimes. Additional details on hyperparameter choice and architectures are provided as supplementary material.

● **Pareto comparison** We analyze pareto efficiency of `hypersolvers` with respect to both ODE ODE solution accuracy and test task classification accuracy. It should be noted that residual fitting does not require task supervision; indeed, test data could be used for `hypersolver` training. Nonetheless, we decide to use only training data for residual fitting, in order to confirm `hypersolver` ability to generalize to unseen initial conditions of the Neural ODE.

*Multiply–accumulate* operations i.e `MACs` are used as a general algorithmic complexity measure. We opt for `MACs` instead of number of function evaluations (NFEs) of the Neural ODE vector field $f_\theta$ since the latter does not take into account computational overheads due to `hypersolver` network $g_\omega$. It should be noted that for these specific architectures, single evaluations of $f_\theta$ and $g_\omega$ correspond to

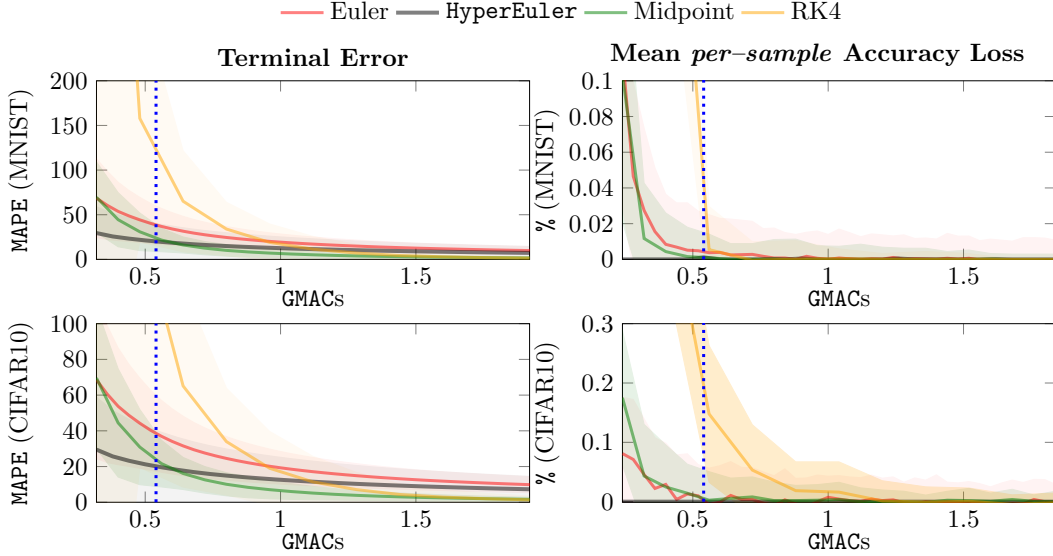

Figure 3: Test accuracy loss %–NFE and MAPE–GMAC Pareto fronts of different ODE solvers on MNIST and CIFAR10 test sets. `HyperEuler` shows higher pareto efficiency for low function evaluations (NFEs) even over higher–order methods.

0.04 `GMACs` and 0.02 `GMACs`, respectively. `HyperEuler` is able to generalize to different step sizes not seen during training, which involved a 10 steps over an integration interval of 1s. Such residual training scheme over 9 residuals corresponds to a computational complexity for `HyperEuler` of 0.54 `GMACs`, highlighted in blue in Fig. 3. As shown in the Figure, `HyperEuler` enjoys pareto optimality over alternative fixed–step methods. The `hypersolver` is able to generalize to different step sizes not seen during training, outperforming higher–order methods such as midpoint and RK4 at low NFEs. As expected, even though higher–order methods eventually surpass `HyperEuler` at higher NFEs as predicted by theoretical bounds, the `hypersolver` retains its pareto optimality over Euler.

⬤ **Wall–clock speedups** We measure wall–clock solution time speedups of various fixed–step methods over `dopri5` for image classification Neural ODEs. Here, absolute time refers to average time across batches of the MNIST test set required to solve the Neural ODE with different numerical schemes.

Each method performs the minimum number of steps to preserve total accuracy loss across the test set to less than 0.1%. As shown in Fig. 4, `HyperEuler` solves an MNIST Neural ODE roughly 8 times faster than `dopri5` and with comparable accuracy, achieving significant speedups even over its base method Euler. Indeed, Euler requires a larger number of steps due to its pareto inefficiency compared to `HyperEuler`, leading to a slower overall solve. The measurements presented are collected on a single `V100` GPU.

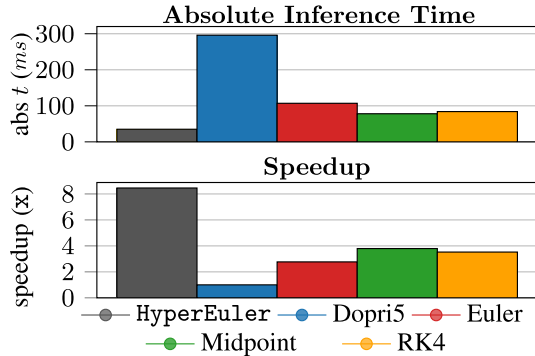

Figure 4: Absolute time ($ms$) speedup of fixed–step methods over `dopri5` (MNIST test set). `HyperEuler` solves the Neural ODE 8x faster than `dopri5` with the same accuracy.

| General Butcher Tableau | | Butcher Tableau of 2nd–order $\alpha$ family | |
|---|---|---|---|
| $\mathbf{c}$ | $\mathbf{A}$ | $0$ | |
| | | $\alpha$ | $\alpha$ |
| | $\mathbf{b}$ | | $1 - \frac{1}{2\alpha}$ $\quad$ $\frac{1}{2\alpha}$ |

Figure 5: *Butcher Tableau* collecting coefficients of numerical methods see e.g (3). [left] general case. [Right] tableau of second–order $\alpha$ family. Note that $\alpha = 0.5$ recovers the *midpoint* method.

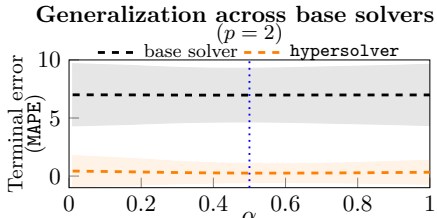

Figure 6: Neural ODE MAPE terminal solution error of `HyperMidpoint` and various members of the $\alpha$–family.

● **Generalization across base solvers** We verify `hypersolver` capability to generalize across different base solvers of the same order. We consider the general family of second–order explicit methods parametrized by $\alpha$ (Süli & Mayers, 2003) as shown in Fig. 5. Employing a parametrizing family for second–order methods instead of specific instances such as midpoint or Heun allows for an analysis of gradual generalization performance as $\alpha$ is tuned away from its value corresponding to the chosen base solver. In particular we consider as midpoint, recovered by $\alpha = 0.5$, as the base solver for the corresponding `hypersolver`.

Fig. 6 shows average terminal MAPE solution error of MNIST Neural ODEs solved with both various $\alpha$ methods as well as a single `HyperMidpoint`. As with the previous experiments, the error is computed over `dopri5` solutions, and averaged across test data batches. `HyperMidpoint` is then evaluated, without finetuning, by swapping its base solver with other members of the $\alpha$ family. The `hypersolver` generalizes to different base solvers, preserving its pareto efficiency over the entire $\alpha$–family.

## 4.2 Lightweight Density Estimation

We consider sampling in the FFJORD (Grathwohl et al., 2018) variant of *continuous normalizing flows* (Chen et al., 2018) as an additional task to showcase `hypersolver` performance. We train CNFs closely following the setup of Grathwohl et al. (2018). Then, we optimize two–layer, second–order Heun hypersolvers, `HyperHeun`, with $K = 1$ residuals obtained against `dopri5` with absolute tolerance $10^{-5}$ and relative tolerance $10^{-5}$. The striking result highlighted in Fig. 7 is that with as little as two NFEs, `Hypersolved` CNFs provide samples that are as accurate as those obtained through the much more computationally expensive `dopri5`.

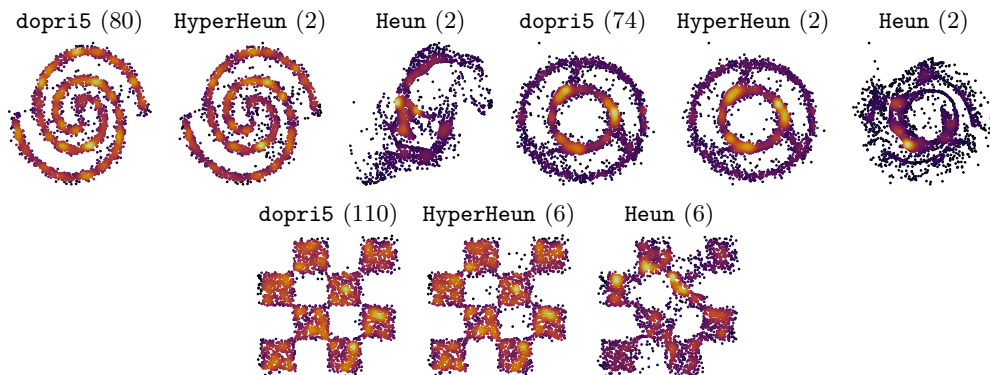

Figure 7: Reconstructed densities with continuous normalizing flows and Heun `hypersolver` `HyperHeun`. The inference accuracy of dopri5 is reached through the `hypersolver` with a significant speedup in terms of computation time and accuracy. Heun method fails to solve correctly the ODE with same NFEs of `HyperHeun`.

## 5 Related Work

**Neural network solvers** There is a long line of research leveraging the universal approximation capabilities of neural networks for solving differential equations. A recurrent theme of the existing work (Lagaris et al., 1997, 1998; Li-ying et al., 2007; Li & Li, 2013; Mall & Chakraverty, 2013; Raissi et al., 2018; Qin et al., 2019) is direct utilization of noiseless analytical solutions and evaluations in

low dimensional settings. Application specific attempts (Xing & McCue, 2010; Breen et al., 2019; Fang et al., 2020) provide empirical evidence in support of the earlier work, though the approximation task is still cast as a gradient–matching regression problem on noiseless labels. Deep neural network base solvers have also been used in the distributed parameters setting for PDEs (Han et al., 2018; Magill et al., 2018; Weinan & Yu, 2018; Raissi, 2018; Piscopo et al., 2019; Both et al., 2019; Khoo & Ying, 2019; Winovich et al., 2019; Raissi et al., 2019). Techniques to use neural networks for fast simulation of physical systems have been explored in (Grzeszczuk et al., 1998; James & Fatahalian, 2003; Sanchez-Gonzalez et al., 2020). More recent advances involving symbolic regressions include (Winovich et al., 2019; Regazzoni et al., 2019; Long et al., 2019).

The hypersolver approach is different in several key aspects. To the best knowledge of the authors, this represents the first example where neural network solvers show both *consistent* and *significant* pareto efficiency improvements over traditional solvers in high–dimensional settings. The performance advantages are demonstrated in the absence of analytic solutions and are supported by theoretical guarantees, ultimately yielding large inference speedups of practical relevance for Neural ODEs.

**Multi–stage residual networks**   After seminal research (Sonoda & Murata, 2017; Lu et al., 2017; Chang et al., 2017; Hauser & Ray, 2017; Chen et al., 2018) uncovered and strengthened the connection bewteen ResNets and ODE discretizations, a variety of architecture and objective specific adjustments have been made to the vanilla formulation. The above allow, for example, to accomodate irregular observations in sequence data (Demeester, 2019) or inherit beneficial properties from the corresponding numerical methods (Zhu et al., 2018). Although these approaches share some structural similarities with the Hypersolved formulation (4), the objective is drastically different. Indeed, such models are optimized for task–specific metrics without concern about preserving ODE properties, or developing a synergistic connection between model and solver.

# 6   Discussion

Hypersolvers can be leveraged beyond the inference step of continuous–depth models. Here, we provide avenues of further development of the framework.

Hypersolver **overhead**   The source of the computational (and memory) overheads caused by the use of hypersolver is indeed represented by the evaluation of $g_\omega$ at each solver step. Nonetheless, this overhead (e.g. in terms of multiply–accumulate operations, MACs) decreases as the solver order increases. In fact, in a $p$th order solver where $f_\theta$ should be evaluated $p$ times, $g_\omega$ is evaluated only once. Let $\text{MAC}_f$, $\text{MAC}_g$ be indicators of algorithmic complexity of $f_\theta$ and $g_\omega$, respectively. We have that the *relative overhead* (in terms of MACs) $\text{O}_r$ is

$$\text{O}_r = \frac{p\text{MAC}_f + \text{MAC}_g}{p\text{MAC}_f} = 1 + \frac{1}{p}\frac{\text{MAC}_g}{\text{MAC}_f}$$

and $\text{O}_r \to 1$ for $p \to \infty$ Thus, the experiments on pareto efficiency and wall–clock speedup using HyperEuler showcased in Sec. 4.1 should be regarded as worst–case scenario, i.e. the most expensive computational–wise.

> Even in the worst-case scenario, hypersolvers remain pareto efficient over traditional methods

**Beyond fixed–step explicit** hypersolvers   In this work, we focus on developing hypersolvers as enhancements to fixed–step explicit methods for Neural ODEs. Although this approach is already effective during inference, hypersolvers are not constrained to this setting. Indeed, the proposed framework can be used to systematically blend learning models and numerical solvers beyond the fixed–step, explicit case. In principle, we could employ hypersolvers into *predictor–corrector* scheme where we may learn higher–order terms of either the (explicit) predictor or the (implicit) corrector, effectively reducing the overall truncation error. Similarly, adaptive stepping might be achieved by augmenting, in example, the *Dormand–Prince* (dopri5) scheme. dopri5 uses six NFEs to calculate fourth- and fifth-order Runge–Kutta solutions and obtain the error estimate for step adaptation. Here, we could substitute RK5 with an HyperRK4 and/or train a NN to perform the adaptation given the error estimate.

> `Hypersolver` are not limited to fixed–step explicit base solvers.

**Accelerating Neural ODE training**  Speeding up continuous–depth model training with `hypersolvers` involves additional challenges. In particular, it is necessary to ensure that the `hypersolver` network remains a $O(\delta)$ approximator of residuals $\mathcal{R}$ across training iterations. A theoretical toolkit to tackle such a task may be offered by *continual learning* (Parisi et al., 2019).

Consider the problem of approximating the solution of a Neural ODE at training iteration $t+1$ having optimized the `hypersolver` on flows generated by the model $f_{\theta_t(s)}(\mathbf{x}_t, s, \mathbf{z}(s))$ at the previous training step $t$. This setting involves a certifiably smooth transition between tasks that is directly controlled by the learning rate $\eta$, leading to the following result

**Proposition 1** (Vector field training sensitivity). *Let the model parameters $\theta_t$ be updated according to the gradient-based optimizer step $\theta_{t+1} = \theta_t + \eta\Gamma(\nabla_\theta\mathcal{L}_t)$, $\eta > 0$ to minimize a loss function $\mathcal{L}_t$ and let $f_{\theta_t}$ be Lipsichitz w.r.t. $\theta$. Then,*

$$\forall \mathbf{z} \in \mathbb{R}^{n_z}, \quad \|\Delta f_{\theta_t}(s, \mathbf{x}, \mathbf{z})\|_2 \leq \eta L_\theta \|\Gamma(\nabla_\theta\mathcal{L})\|_2$$

*being $L_\theta$ the Lipschitz constant.*

By leveraging the above result, or pretraining the `hypersolver` on a sufficiently large collection of dynamics, it might be possible to construct a training procedure for Neural ODEs which maximizes `hypersolver` reuse across training iterations. Similar to other application areas such as language processing (Howard & Ruder, 2018; Devlin et al., 2018), we envision pretraining techniques to play a fundamental part in the search for easy–to–train continuous–depth models.

> Maximizing `hypersolver` reuse represents an important objective for faster Neural ODE training.

**Model–solver joint optimization**  `Hypersolver` and Neural ODE training can be carried out *jointly* during optimization for the main task. Beyond numerical accuracy metrics, other task specific losses can be considered for `hypersolvers`. In the standard setting, numerical solvers act as adversaries preserving the ODE solution accuracy at the cost of expressivity. Taking this analogy further, we propose adversarial optimization in the form $\min_\omega \max_\theta \sum_{k=0}^{K} \|\mathbf{z}_k - \bar{\mathbf{z}}_k\|_2$ where $\bar{\mathbf{z}}_k$ is the solution at mesh point $k$ given by an adaptive step solver. When used either during hypersolver pretraining or as a regularization term for the main task, the above gives rise to emerging behaviors in the dynamics $f_{\theta(s)}$ which exploit solver weaknesses. We observe, as briefly discussed in the Appendix, that direct adversarial training teaches $f_{\theta(s)}$ to leverage *stiffness* (Shampine, 2018) of the differential equation to increase the `hypersolver` solution error.

> Adversarial training may be used to enhance `hypersolver` resilience to challenging dynamics.

## 7   Conclusion

Computational overheads represent a great obstacle for the utilization of continuous–depth models in large scale or real–time applications. This work develops the novel `hypersolver` framework, designed to alleviate performance limitations by leveraging the key model–solver interplay of continuous–depth architectures. `Hypersolvers`, neural networks trained to solve Neural ODEs accurately and with low overhead, improve solution accuracy at a negligible computational cost, ultimately improving pareto efficiency of traditional methods. Indeed, the synergistic combinations of `Hypersolvers` and Neural ODEs enjoy large speedups during inference steps of standard benchmarks of continuous–depth models, allowing in example accurate sampling from *continuous normalizing flows* (CNFs) in as little as 2 *number of function evaluations* (NFEs). Finally, we discuss how the `hypesolver` paradigm can be extended to enhance Neural ODE training through continual learning, pretraining or joint optimization of model and `hypersolver`.

## Broader Impact

Major application areas for continuous deep learning architectures so far have been generative modeling (Grathwohl et al., 2018) and forecasting, particularly in the context of patient medical data (Jia & Benson, 2019). While these models have an intrinsic interpretability advantages over discrete counterparts, it is important that future iterations preserve these properties in the search for greater scalability. Early adoption of the hypersolver paradigm would speed up widespread utilization of Neural ODEs in these domains, ultimately leading to positive impact in healthcare applications.

Accurate forecasting is at the foundation of system identification and control, two additional application areas set to be greatly impacted by continuous models. Unfortunately, theoretical guarantees of robustness in the worst–case scenario are challenging to construct for data–driven approaches. As these approaches are refined, they are also likely to negatively impact the employment market by accelerating job automation in critical areas.

## Acknowledgment

We thank Patrick Kidger for helpful discussions. This work was supported by the Korea Agency for Infrastructure Technology Advancement (KAIA) grant, funded by the Ministry of Land, Infrastructure and Transport under Grant 19PIYR-B153277-01.

## Footnotes

[2]Compared with an equivalent–performance ResNet.

[3]Numerical solvers which obey to (2) are called *explicit* solvers

[4]Supporting reproducibility code is at

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
