[Supplementary Material]

# Hypersolvers: Toward Fast Continuous–Depth Models
## *Supplementary Material*

## Table of Contents

## A   Theoretical Results

### A.1   Proof of Theorem 1

**Theorem 1** (Hypersolver Local Truncation Error). *If $g_\omega$ is a $\mathcal{O}(\delta)$ approximator of $\mathcal{R}$, i.e.*
$$\forall k \in \mathbb{N}_{\leq K} \quad \|\mathcal{R}(s_k, \mathbf{z}(s_k), \mathbf{z}(s_{k+1}) - g_\theta(\epsilon, s_k, \mathbf{x}, \mathbf{z}(s_k))\|_2 \leq \mathcal{O}(\delta),$$
*then, the local truncation error $e_k$ of the* `hypersolver` *is* $\mathcal{O}(\delta\epsilon^{p+1})$.

*Proof.* We can directly compute the local truncation error for the `hypersolver` as
$$e_k = \|z(s_{k+1}) - z(s_k) - \epsilon\psi(s_k, x, z(s_k)) - \epsilon^{p+1}g_\omega(\epsilon, s_k, x, z(s_k))\|_2$$
Thus,
$$e_k = \epsilon^{p+1}\|\mathcal{R}(s_k, z(s_k), z(s_{k+1})) - g_\omega(\epsilon, s_k, x, z(s_k))\|_2$$
$$\leq \mathcal{O}(\delta\epsilon^{p+1})$$
□

### A.2   Proof of Proposition 1

**Proposition 1** (Vector field training sensitivity). *Let the model parameters $\theta_t$ be updated according to the gradient-based optimizer step $\theta_{t+1} = \theta_t + \eta\Gamma(\nabla_\theta\mathcal{L}_t)$, $\eta > 0$ to minimize a loss function $\mathcal{L}_t$ and let $f_{\theta_t}$ be Lipsichitz w.r.t. $\theta$. Then,*
$$\forall \mathbf{z} \in \mathbb{R}^{n_z}, \quad \|\Delta f_{\theta_t}(s, \mathbf{x}, \mathbf{z})\|_2 \leq \eta L_\theta \|\Gamma(\nabla_\theta\mathcal{L})\|_2$$
*being $L_\theta$ the Lipschitz constant.*

*Proof.* For the Lipschitz continuity of $f_\theta$, it holds
$$\forall\theta, \theta' \in \mathbb{R}^{n_\theta} \quad \|f_\theta - f_{\theta'}\|_2 \leq L_\theta\|\theta - \theta'\|_2$$
Thus,
$$\|\Delta f_{\theta_t}(x)\|_2 := \|f_{\theta_{t+1}}(x) - f_{\theta_t}(x)\|_2 \leq L_\theta\|\theta_{t+1} - \theta_t\|_2 = \eta L_\theta\|\Gamma(\nabla_\theta\mathcal{L})\|_2$$
□

# B  Further Discussion

## B.1  Software implementation

We provide PyTorch (Paszke et al., 2017) code showcasing a general `hypersolver` template:

```python
class HyperSolver(HyperSolverTemplate):
    def __init__(self, f, g, base_solver):
        super().__init__(f, g)
        self.base_solver = base_solver
        self.p = self.base_solver.order

    def forward(self, ds, dz, z):
        "Calculates single residual"
        ds = ds*torch.ones([*z.shape[:1], 1 , *z.shape[2:]]).to(z)
        z = torch.cat([z, dz, ds], 1)
        z = self.g(z)
        return z

    def base_residuals(self, base_traj, s_span):
        "Computes residuals of `base_solver` on `base_traj`"
        ds = s_span[1] - s_span[0]
        fi = torch.cat([self.base_solver(s, base_traj[i], self.f, ds)[None,:,:]
                        for i, s in enumerate(s_span[:-1])
                       ])
        return (base_traj[1:] - base_traj[:-1] - ds*fi)/ds**(self.p + 1)

    def hypersolver_residuals(self, base_traj, s_span):
        "Applies the hypersolver on `base_traj` to compute residuals"
        ds = (s_span[1] - s_span[0]).expand(*base_traj[:-1].shape)
        dz = torch.cat([self.f(s, base_traj[i])[None,:,:]
                        for i, s in enumerate(s_span[:-1])
                       ])
        residuals = torch.cat([self(ds_[0,0], dz_, z_)[None]
                               for ds_, dz_, z_ in zip(ds, dz, base_traj[:-1])
                              ], 0)
        return residuals

    def odeint(self, s_span, z, use_residual=True):
        "Solves the ODE in `s_span`"
        traj = torch.zeros(len(s_span), *z.shape); traj[0] = z
        for i, s in enumerate(s_span[:-1]):
            ds = s_span[i+1] - s_span[i]
            dz = self.f(s, z)
            if use_residual: z = z + ds*self.base_solver(s, z, self.f, ds)
                                + ds**(self.p + 1) * self(ds, dz, z)
            else: z = z + ds*self.base_solver(s, z, self.f, ds)
            traj[i+1] = z
        return traj
```

## B.2  Adversarial training

*Stiffness* in differential equations is an important problem of practical relevance as it often requires development of specialized solution methods (Shampine & Gear, 1979; Cash, 2003). While challenging to fully characterize, stiffness occurs when adaptive–step solvers require a high number of solution steps to maintain the error below specified tolerances, in regions where the solution appears otherwise relatively smooth. Indeed, stiff ODEs are generally difficult to solve accurately for fixed–step solvers. Direct adversarial training allows $f_{\theta(s)}$ to find and exploit common weaknesses of numerical methods, which in turn improves `hypersolver` resilience to a wider class of dynamics.

Figure 8: Pareto comparison of different solvers in the *trajectory tracking* task.

Figure 9: MAPE–NFE pareto fronts of different ODE solvers on MNIST and CIFAR10 test sets. `HyperEuler` shows higher pareto efficiency for low function evaluations (NFEs) even over higher–order methods.

## C    Experimental Details

**Computational resources**    The experiments have been carried out on a machine equipped with a single `NVIDIA Tesla V100` GPU and an eight–core Intel Xeon processor. In addition, we measure *wall–clock* speedups on a few additional hardware setups and found the results to be consistent.

### C.1    Additional Experiments

**Trajectory tracking**    To evaluate the effectiveness of the trajectory fitting method, we consider a Galërkin Neural ODE (Massaroli et al., 2020b) tasked to tracking of a periodic signal $\beta(s)$. The Neural ODE is optimized with an integral loss of the type $(\mathbf{z}(s) - \beta(s))^2$ in the integration domain $S := [0, 1]$. After the initial training of the model, we fit a three–layer `HyperEuler` of hidden dimensions $64, 64, 64$ using a *trajectory* fitting approach.

Fig. 8 shows that the pareto efficiency in terms of global truncation error $\mathcal{E}(k)$ is preserved when training with *trajectory fitting*. In the 10 - 25 NFE range, `HyperEuler` results more efficient than higher–order solvers such as midpoint and RK4.

### C.2    Image Classification

We report a detailed discussion on the hyperparameter and architectural choices made for the image classification experiments. Further pareto efficiency experimental results, measured in NFEs instead of `MACs`, are provided in Fig. 9. We omit test accuracy loss NFE pareto fronts since `hypersolvers` avoid test accuracy losses altogether as shown in the main text.

**Training hyperparameters**    On MNIST, we optimized Neural ODEs for 20 epochs with batch size 32 utilizing the Adam optimizer with learning rate $3^{-3}$ and a cosine annealing scheduler down to $10^{-4}$ at the end of training. On CIFAR10, we utilized a similar strategy, with 20 epochs, batch size 32 and the same optimizer.

The `HyperEuler` hypersolver has been trained utilizing fitting the residuals of the *Dormand–Prince* solver (`dopri5`) (Dormand & Prince, 1980) with absolute and relative tolerances set to $10^{-4}$. We use the AdamW (Loshchilov & Hutter, 2017) optimizer with $lr = 10^{-2}$ and a cosine annealing schedule down to $5 * 10^{-4}$.

The `hypersolver` training is subdivided into two phases, proceeding as follows. First, we stabilize the optimization by pretraining the `hypersolver` on the trajectories generated from a single batch for several iterations, usually 10. After this initial phase, the data batch is swapped every 10 iterations. This allows the `hypersolver` to generalize by having access to trajectories generated from different batches of the training set.

We experimented with different numbers of iterations for `hypersolver` training. Convergence has been observed in as quickly as 5000 iterations, corresponding to *less than* 3 epochs of the MNIST training dataset with batch size 32. In practice, 15000 iterations (or 10 epochs) is sufficient to produce results comparable to the ones shown in Figure 3. A similar discussion applies to CIFAR10.

**Architectural details**   In the following, we report `PyTorch` code defining the Neural ODE and `hypersolver` architectures in full. The code snippets are followed by a text description for accessibility. In MNIST, the architecture takes the form

```
f = nn.Sequential(nn.Conv2d(32, 46, 3, padding=1),
                  nn.Softplus(),
                  nn.Conv2d(46, 46, 3, padding=1),
                  nn.Softplus(),
                  nn.Conv2d(46, 32, 3, padding=1))

nde = NeuralODE(f)

model = nn.Sequential(nn.BatchNorm2d(1),
                      Augmenter(augment_func=nn.Conv2d(1, 31, 3, padding=1)),
                      nde,
                      nn.AvgPool2d(28),
                      nn.Flatten(),
                      nn.Linear(32, 10))
```

where the input–augmented layer (Massaroli et al., 2020b) Neural ODE $f_\theta$ is defined as a sequence of convolutional layers of channel dimensions $12, 64, 12$ and kernel size 3. The complete architecture is then composed of the above defined Neural ODE with a deconvolution layer, and a linear fully–connected layer to output the classification probabilities.

The `HyperEuler` architecture $g_\omega$ is simpler and is composed of only a two–layer CNN with parametric–ReLU (PReLU) (He et al., 2015) activation. The input layer channel dimension is 25 whereas the input to $f_\theta$, $z(0)$ is only augmented to 12 channels. This is because $g_\omega$ takes a concatenation of $z, f_\theta(z), s$ which yields $12 + 12 + 1$ channels.

```
g = nn.Sequential(nn.Conv2d(32+32+1, 32, 3, stride=1, padding=1),
                  nn.PReLU(),
                  nn.Conv2d(32, 32, 3, padding=1),
                  nn.PReLU(),
                  nn.Conv2d(32, 32, 3, padding=1))
```

For the CIFAR10 experiments, on the other hand, $f_\theta$ and the complete architectures are defined as

```
f = nn.Sequential(nn.Conv2d(20, 50, 3, padding=1),
                  nn.Softplus(),
                  nn.Conv2d(50, 50, 3, padding=1),
                  nn.Softplus(),
                  nn.Conv2d(50, 20, 3, padding=1))

nde = NeuralODE(f)
```

```
model = nn.Sequential(nn.BatchNorm2d(3),
                      nn.Conv2d(3, 20, 3, padding=1),
                      nde,
                      nn.AdaptiveAvgPool2d(2),
                      nn.Flatten(),
                      nn.Linear(20*4, 10))
```

The tt HyperEuler architecture is

```
g = nn.Sequential(nn.Conv2d(20+20+1, 32, 3, padding=1),
                  nn.PReLU(),
                  nn.Conv2d(32, 32, 3, padding=1),
                  nn.PReLU(),
                  nn.Conv2d(32, 20, 3, padding=1))
```

It should be noted that even though the Neural ODEs achieve comparable results as (Dupont et al., 2019; Massaroli et al., 2020b), the focus of these experiments has not been optimizing $f_\theta$ for task–performance. Indeed, we observed that `HyperEuler` obtains similar results to those shown in the main body of the paper and in Figures 3 and 9 across a variety of different $f_\theta$. The setup for base solver generalization experiments has been the same as MNIST experiments, with the only major difference being a choice of `HyperMidpoint` and an evaluation across different base solvers.

**Results**   To highlight the efficacy of `hypersolvers`, we utilize the following metrics

- *Absolute error* of the numerical solution at different solution mesh points. These results provide qualitative proof of the higher solution accuracy of `hypersolvers` across different types of data samples.

- *Mean absolute percentage error* (MAPE) of the terminal solution. Pareto efficiency of `hypersolver` numerical solutions.

- *Average test accuracy decrement*. We measure the average (across samples) accuracy lost by a transition away from `dopri5`. The objective has been to show that outside of solution accuracy, `hypersolvers` offer pareto efficiency over other solvers in terms of task–specific metrics.

### C.3   Continuous Normalizing Flows

We optimize *continuous normalizing flows* (CNF) (Chen et al., 2018) on density estimation tasks, closely following the setup of (Grathwohl et al., 2018). For a complete reference on normalizing flows we refer to (Kobyzev et al., 2019).

In particular, the training for the two–dimensional tasks is carried out for 3000 iterations with an Adam optimizer set to constant learning rate $10^{-3}$. The CNF is constructed with a three–layer MLP of hidden dimensions 128, 128, 128 and the corresponding ODE is solved with `dopri5` with absolute and relative tolerances set to $10^{-5}$ for an accurate forward propagation of the log–density change (Chen et al., 2018). We consider several standard two–dimensional densities following (Grathwohl et al., 2018), namely `pinwheel`, `rings`, `checkerboard` and a modified, more challenging `circles` where the annuli are connected by three curves.

After this initial step, we train an *Heun hypersolver* for 30000 iterations of residual fitting on backward trajectories utilizing a similar strategy as discussed in the previous subsection. Namely, we leverage *AdamW* (Loshchilov & Hutter, 2017) with $lr = 5^{-3}$, weight decay $10^{-6}$ and a two–stage training where the data–sample generating the residuals is switched after every 100 iterations.