[Reviews · NeurIPS 2020]

Review 1

Summary and Contributions: In this paper, the authors propose the hypersolvers. Given a solution to a neural ODE by an solve with low tolerance, another neural network is learned to approximate the residual, thus improves the truncation error. The proposed idea seems interesting and, to the best of my knowledge, novel. The authors empirically demonstrate the improvement in the Pareto frontier for accuracy and computational efficiency over existing solvers. **post-rebuttal**: I have read the authors' rebuttal and would like to keep my rating.

Strengths: I believe this work will be of interest to the community, and the idea seems interesting and novel.

Weaknesses: The experimental study seems not strong enough. I am curious why the authors only conduct experiments on FFJORD sampling, but not likelihood evaluations. Furthermore, different aspects discussed in Section 6 are interesting, but it would make the paper stronger if the authors further explore those ideas.

Correctness: The claims and method look correct to me.

Clarity: The paper is well organized, clearly written and easy to read.

Relation to Prior Work: The authors sufficiently discussed the related work and how this work differs from them.

Reproducibility: Yes

Additional Feedback: I was wondering how deep this proposed method can go, like a hyper-hypersolver, using another network to fit the residue of the hypersolver. This might lead to a progressive training of an ensemble of networks? Typo: line 228 on page 8, Fig 6 should be Fig 7?


Review 2

Summary and Contributions: This paper proposes to use a neural net to learn the high-order terms in the local Taylor expansion of a variable governed by an ODE, and utilizes this learned information to speed up neural ODE inference.

Strengths: The paper proposes a simple idea that seems to be effective. The problem of neural ODE inference is important but rather less studied by the community.

Weaknesses: There are certain aspects of the method that are not showcased as extensively as they should. I detail these in the Additional feedback section of this review.

Correctness: Claims and methods are correct to the best of my knowledge.

Clarity: The paper is well-written to the best of my knowledge.

Relation to Prior Work: Relation is clear to the best of my knowledge.

Reproducibility: Yes

Additional Feedback: - An important aspect of inference is the actual time and memory cost. While this is obviously dependent on the architecture of the computer, hardware, and software implementation, the overall message of inference speedup would be more convincing if at least some statistics/plots were given on this (e.g. on a computer w/ 12 CPU cores and a Nvidia v100 with PyTorch version 1.2.0). - Implicitly, there's a hyper-parameter to be tuned, which is the tolerance level set for obtaining training data for the high-order term neural net. How sensitive is the method w.r.t. this parameter? - Figure 5 top right plot's xticks and xlabel seems to be overlapping, making it hard to read. Post-rebuttal: I am satisfied with the additional experiments demonstrating the speed advantage of this method in practice, and therefore raising my score to a 7.


Review 3

Summary and Contributions: The paper presents a method to increase solver accuracy in Neural ODE models by fitting a residual neural network for a given numerical method. Thus, online computational cost of the numerical solver can be preprocessed in an offline neural network training phase. The quality of the achieved numerical fit is evaluated.

Strengths: The paper highlights an important venue for further research in Neural ODEs.

Weaknesses: There are two major weaknesses with this submission: 1.) The presented methodology boils down to the following idea: pre-computing necessary numerical methods offline, feed the results into a fast-to-evaluate interpolator and use the interpolator in deployment for faster prediction times. Stated this broadly, the idea is not novel and has many names in different communities, e.g., surrogate models, white-box emulation and similar. Thus, the paper would not need to convince me that this is generally a good idea, but why /this particular interpolation of the paper/ is better suited than the many alternatives, e.g., simply precomputing a mesh of possible IVPs with high accuracy and training directly g : x |-> z(s_K). The discussion in Sect. 6 tends towards this rationale, but is too short and superficial to be of substantial benefit. 2.) In particular, the authors present a rationale for supposedly faster evaluation times using Hypersolvers, but there are no experiments whatsoever what sort of hypersolver-training time/accuracy tradeoff is required.

Correctness: The theoretical claims seem to be correct but I have only checked the superficially. The paper's discussion and experiments fail to address why this particular form of numerical pre-computation and interpolation is beneficial. The experiments that are presented are reasonable, but do not support the main story of the paper.

Clarity: The paper is well written.

Relation to Prior Work: The connections to other work in Neural ODEs are adequately addressed. The connection to surrogate models is entirely absent. Some examples include: http://www.jmlr.org/papers/volume11/gorissen10a/gorissen10a https://dl.acm.org/doi/abs/10.1145/882262.882359 https://ieeexplore.ieee.org/document/7469355?arnumber=7469355 https://dl.acm.org/doi/pdf/10.1145/280814.280816

Reproducibility: Yes

Additional Feedback: Post-rebuttal update: Unfortunately, I have to report that the authors' comments do not change my evaluation as I have the impression that they miss my point: I fully agree with the authors that Sect. 6 contains interesting hypotheses that I would like to see published at NeurIPS---if the hypotheses are proven to be correct and the authors can get the methods to work. So far, they are merely thought-provoking speculations and thus do not support the publication at this point per se. Secondly, I feel that the points "Scope" and "Alternative Approaches" cannot be seperated in this case. I also agree with the authors that they do not claim to solve a more general problem. I also agree with the authors that a clever idea for a tailored recipe in the context of Neural ODEs could be interesting. However, I remain unconvinced that *this particular solution* is specifically beneficial in *this particular context*. Specially so, as I claim that from the experiments I cannot make a comparison with other methods (e.g., directly predicting final time). Doubly so, as fixed step methods are not ODE solvers, i.e., there are no checks whatsoever within the solver to check for any sort of numerical accuracy. If I have understood the paper correctly, the training data for the Hypersolver was generated under the same solver and configuration as has later been used for testing. This is not a problem per se, but it is not obvious why the solution from this solver-configuration-pair should be the gold standard in the application context. I have the impression that the metric punishes the Midpoint and other methods for not being Dorpi5 at RelTol/AbsTol 1e-5. Furthermore, I want to highlight something else about the proposed method: it does *not necessarily* learn *the next higher-order* error term. It really learns *the difference* between any two numerical methods. If the authors would have trained the Hypersolver on Dopri5 RelTol/AbsTol 1e-10, the authors probably would have gotten a numerical method that behaved like a Dopri5 RelTol/AbsTol 1e-10. The fact that the Euler's method is still applied only acts as a determinstic pre-computation per step that the Hypersolver can correct for given a large enough model class (and, as I understand the experiments, also does). This is the particular reason why I would like to see experiments that compare with a) simply predicting the next step (without additional Euler computations) or b) simply predicting the final solution. I agree with the authors that there are probably interesting accuracy/complexity trade-offs, but from the current set of experiments, I am unable to judge them. I hope the following more concrete suggestions helps the authors in their future submissions: 1) The authors need to show that the effect of the Hypersolver is not due to overfitting to solver-configuration pairs. In particular, for testing either a different solver with same tolerance settings or the same solver with much lower tolerance settings should be used. 2) In this context, also test the effect of combining a hypersolver with *different* numerical schemes of the *same* order. (E.g., Ralston's method and Heun's method). If the positive effect of the Hypersolver really is due to predicting the next leading order term, this should work reasonably well. 2b) But also try to simply regress z(s_k+1) from z(s_k) and 2c) z(s_K) from z(s_0). All of these different approaches should work in general and the interesting question is whether any of those show a particularly beneficial accuracy/complexity trade-off. 3) Or, the authors could focus on detailing out the ideas presented in Sect. 6 (possibly a paper each). I personally believe that these would have the much higher impact in the community. I hope this clarification helped in understanding my criticism and also helps the authors to improve their manuscript for future submissions.


Review 4

Summary and Contributions: The authors propose to refine classic time stepping schemes for solving ODEs by modelling the local error with a network.

Strengths: It is empirically shown that the proposed method indeed improves the accuracy of Eules/Heuns method.

Weaknesses: It is not clear how the authors define number of function evaluations (NFE), which is the metric used to claim solver speedups. Thus it is not clear how much is gained in absolute terms. I elaborate on this issue below.

Correctness: Seems so.

Clarity: It is fairly well written.

Relation to Prior Work: Yes.

Reproducibility: Yes

Additional Feedback: Some comments: 1) In footnote 2: single-step and explicit are not synonymous, e.g. implicit Euler is a single-step method but not explicit. I guess "single-step or explicit" should be "explicit single-step". 2) Examining the data fitting criterion \ell it is quite clear that the minimizer is given by g(e,x,s_k,z(s_k) ) = - 1/e^{p+1}*( z(s_{k+1) - z(s_k) - e*\psi ). Consequently, the hypersolver in (4) would reduce to z_{k+1} = z(s_{k+1). So why not just train a NN to map (x,s) to z(s)? This would also reduce the input dimension. 3) \ell_{local} is not defined anywhere. 4) The number of function evaluations (NFE) is not defined. Do you mean number of evaluations of the vector field f? In such a case this might be misleading since these hypersolvers, in addition to using function evaluations for the underlying classic ODE solver also have to evaluate the NN-correction. That is, this would only be a viable metric of computational complexity if the NN-correction is very cheap to compute in comparison to the vector field, is this the case? 5) It is claimed that the hypersolver is able to generalise to different step-sizes. It is not obvious where the support for this claim can be found. Nontheless it makes sense in view of my comment 2).

[Author Response · NeurIPS 2020]

1. We thank the reviewers for thoughtful feedback, and for acknowledging the importance and novelty of this work. We

2. focus on **absolute speedups** and **trade–offs** by providing additional results as further evidence of practical effectiveness.

3. **Time and memory trade–offs of hypersolvers [All]:** We agree that the number of function evaluations of the Neural

4. ODE vector field $f$ (NFE), while a commonly used measure in the literature, does not take into account the hypersolver

5. cost. To address concerns related to hypersolver overheads, Fig. 1 provides additional pareto optimality plots (solution

6. error & test acc. loss) against MACs, *multiply-accumulate* operations of $f$ (for baselines) and $f + g$ (for hypersolvers).

7. MACs, similar to FLOPs, are commonly used as an hardware–agnostic algorithmic complexity measure. We note that in

8. the current implementation, HyperEuler corrections require less than $50\%$ of the MACs required for an evaluation of

9. the $f$ network. In cases where the architecture of $f$ is itself is deeper, the overhead of hypersolvers is reduced, further

10. strengthening their pareto efficiency. Memory overheads are of similar magnitude, though memory is less of a concern

11. in Neural ODE due to their smaller footprint enabled by the $O(1)$ memory adjoint sensitivity technique for training.

12. **Absolute speedup [All]:** To further contextualize the efficacy of hypersolvers, we provide absolute time and speedup

13. plots (TITAN V, CUDA v10.2) in Fig. 2. The baselines are set to perform the minimum number of steps necessary to

14. preserve test classification accuracy. In contrast with MACs, these results take into account implementation and hardware

15. overheads. It should be noted that the relative baseline ranking can different w.r.t MAC plots due to implementation

16. overheads (here torchdiffeq). Here, HyperEuler provides 8x speedup over dopri5 with exact same test accuracy.

17. **Scope [R3]:** We agree with **R3** that there exist a vast literature on speeding up simulators and solvers with neural

18. networks, and have added the references suggested by **R3**. We do not claim to be the first to introduce the idea of

19. offline solver pretraining, nor do we claim state–of–the–art in the general case. The core contribution is comprised

20. of the hypersolver formulation, theoretical guarantees and training strategies tailored for performance in the context

21. continuous models. We believe the emerging learning–based interplay between Neural ODEs and their solver (Sec. 6)

22. to also be a valuable contribution in itself, especially given the increasing use of Neural ODEs across scientific fields.

23. **Alternative approaches and surrogate models [R3, R4]:** We argue that learning only the residual *higher–order* term,

24. instead of the full map $\mathbf{z_k} \mapsto \mathbf{z_{k+1}}$ is advantageous, in the context of Neural ODEs, for several reasons. We notice that

25. the residual fitting requires a substantially smaller NN compared to learning directly the solution of the ODE, which

26. does not make explicit use of the vector field $f$. Th.m 1 further provides theoretical guarantees on the solver's truncation

27. error improvement, relevant for safety–critical applications. Moreover, to the best of our knowledge, this is the first

28. Neural ODE solver designed to learn residuals, showcasing **large speedups** (as shown above) of **practical relevance**.

29. **CNF likelihood eval. [R1]:** Hypersolvers can also be used during CNF likelihood evaluation. We verified this experi-

30. mentally and obtained comparable results to CNF sampling; a discussion will be included.

31. **Extension of Section 6 [R1, R3]:** We agree that Sec. 6 represents an important component of the work and in its

32. current form already highlights original aspects of model–solver interplay. We decided, however, to dedicate more

33. space to the core formulation and results to provide a solid foundation for more complex hypersolver architectures.

34. **Sensitivity to tolerance of ground-truth method [R2]:** Tolerances can be regarded as hyperparameter of $f$ itself. If

35. insufficient, the solutions might have high *error* and lower Neural ODE task performance; regardless, the hypersolver

36. will learn to match the residuals, ensuring the base method is able to track ground–truth solutions. Hypersolvers are

37. only sensitive to tolerances in the sense that they represent an upper–bound on the accuracy of *hypersolved* solutions.

38. **Generalization to different step–sizes [R4]:** Generalization across step–sizes is shown via pareto plots (also Fig. 1).

39. The integration interval is fixed to $S := [0, 1]$ for all NFEs; hence, traversing the *x–axis* corresponds to a denser

40. discretization of $S$. HyperEuler is competitive even far from its training step size $\epsilon = 0.1$ (10 NFEs).

41. **Hyper–hyper [R1]:** This is a valuable suggestion. By progressively training an ensemble networks on increasing–order

42. residuals one could reach a local truncation err. $\mathcal{O}(\delta_1 \delta_2 \cdots \delta_p \epsilon^2)$ instead of $\mathcal{O}(\delta \epsilon^{p+1})$ obtained by directly fitting the

43. residual of a $p$-ord. solver. Understanding whether this is convenient in practice is certainly worth further investigations.

44. **Clarifications:** [R4] $\ell_{\widehat{\texttt{local}}} \to \ell$

Figure 1: Test acc. loss (MNIST) and solution MAPE (avg. across batches) w.r.t MACs. $10^9$ MACs := 1GMAC. HyperEuler always shows higher pareto optimality. Note that 0.4 GMACs $\approx$ cost of 1 eval. of vector field $f$, 0.2 GMACs $\approx$ an eval. of $g$.

Figure 2: Absolute time (ms) and speedup, MNIST test set inference (avg. across data–batches) of various discretization schemes w.r.t the ground–truth solver dopri5. All fixed–step methods perform the minimum number of steps to preserve *total* test set classification accuracy loss to $< 0.1\%$.

[Meta-Review · NeurIPS 2020]

The author feedback caused the reviewers to re-evaluate some of their initial opinions. R2 increased their score, and R3 (who is an expert) carefully reevaluated their opinion, although they did not change their score. (Since the authors speculated about their situation, I want to add that this reviewer is not in any evident conflict of interest, nor are they the author of any related works they pointed out). I largely follow the argument of R3. I understand that the authors are unhappy with the way this review is phrased, but it does raise important issues: There are a lot of design decisions here, not all of which can just be swept under the rug. Of course the proposed method is specifically designed for the application in the NODE setting, and that's fine. However, as R3 rightly points out, the experiments are done in a way that does not allow any generalization beyond a specific combination of network and solver. As we are still in the early days of NODE research, it seems unlikely that even interested readers would be willing to commit themselves to such specific choices only to achieve the speedup reported in the rebuttal. Nevertheless, the other reviewers, perhaps rightly, point out that the paper does contain interesting results that should be published. Indeed, we *are* in the early days of NODE research, and there is scope to test out ideas even if their eventual utility is still a bit unclear. The paper can thus be accepted. However, in light of what I wrote above and the criticism by R3, I want to *strongly urge* the authors to clarify and expand the scope of what we as a community can learn from the paper. In particular, how can the results shown in this paper be transferred to other base solvers?